# Motor Cortex Inhibition and Facilitation Correlates with Fibromyalgia Compensatory Mechanisms and Pain: A Cross-Sectional Study

**DOI:** 10.3390/biomedicines11061543

**Published:** 2023-05-26

**Authors:** Kevin Pacheco-Barrios, Danielle Carolina Pimenta, Anne Victorio Pessotto, Felipe Fregni

**Affiliations:** 1Neuromodulation Center and Center for Clinical Research Learning, Spaulding Rehabilitation Hospital, Massachusetts General Hospital, Harvard Medical School, Boston, MA 02129, USA; kevin.pacheco.barrios@gmail.com (K.P.-B.); dcdasilva@partners.org (D.C.P.); avictoriopessotto@partners.org (A.V.P.); 2Vicerrectorado de Investigación, Unidad de Investigación para la Generación y Síntesis de Evidencias en Salud, Universidad San Ignacio de Loyola, Lima 15024, Peru; 3Laboratório de Imunohematologia e Hematologia Forense (LIM40), Departamento de Medicina Legal, Ética Médica e Medicina Social e do Trabalho, Hospital das Clínicas da Faculdade de Medicina da Universidade de São Paulo (HC da FMUSP), São Paulo 05403-010, Brazil

**Keywords:** fibromyalgia, transcranial magnetic stimulation, intracortical inhibition, intracortical facilitation

## Abstract

The role of transcranial magnetic stimulation (TMS) measures as biomarkers of fibromyalgia syndrome (FMS) phenotypes is still unclear. We aimed to determine the clinical correlates of TMS measures in FMS patients. We conducted a cross-sectional analysis that included 58 patients. We performed standardized TMS assessments, including resting motor threshold (MT), motor-evoked potential (MEP), short intracortical inhibition (SICI), and intracortical facilitation (ICF). Sociodemographic, clinical questionnaires, and quantitative sensory testing were collected from all of the patients. Univariate and multivariate linear regression models were built to explore TMS-associated factors. We found that SICI did not significantly correlate with pain levels but was associated with sleepiness, comorbidities, disease duration, and anxiety. On the other hand, ICF showed a positive correlation with pain levels and a negative correlation with body mass index (BMI). BMI was a negative effect modifier of the ICF and pain association. The clinical correlates of MT and MEP were scarce. Our results suggest that SICI and ICF metrics are potential phenotyping biomarkers in FMS related to disease compensation and levels of pain perception, respectively. The clinical translation of TMS paired-pulse protocols represents an opportunity for a mechanistic understanding of FMS and the future development of precision treatments.

## 1. Introduction

Fibromyalgia syndrome (FMS) is characterized by widespread musculoskeletal pain, fatigue, sleep disturbances, and even autonomic and cognitive dysfunctions [1,2]. FMS affects around 2% of the general population; however, in patients with comorbidities or in hemodialysis, this figure could be as high as 4% or up to 80% in patients with Behcet syndrome [3]. These comorbidities include psychiatric disorders and other chronic pain conditions [4]. Due to this variable clinical presentation, FMS is usually overlooked and underdiagnosed [2,5], and the treatment remains challenging and complex with limited therapeutic efficacy [5,6,7].

FMS represents an umbrella term for multiple clinical phenotypes. Some patients show a main neuropathic component (small-fiber neuropathy), even from potential autoimmune origin [8,9,10], while others show a predominant affected endogenous pain modulation system (EPMS), potentially involving serotoninergic and endogenous opioidergic pathways [11,12], and some present a combination of mechanisms that are hard to disentangle [13]. Identifying and characterizing potential phenotypes could optimize diagnosis, management, and prognosis for FMS patients [14].

Consequently, there is a need for biomarkers in FMS to objectively identify clinical phenotypes and to understand its physiopathological mechanisms [15,16]. Central sensitization seems to play a major role in the mechanisms of FMS [1,17,18,19]. Hyperalgesia is present in most FMS patients and depends almost exclusively on central pain mechanisms, which supports this hypothesis [18]. However, the origin and pathological pathways behind central sensitization in FMS are not clear. Several mechanisms seem to correlate, such as a lack of inhibitory control in pain-related networks (sensorimotor, pre-frontal, and limbic areas) [5], neuroendocrine system [20], neurotransmitters [21], and immunological dysfunctions [22].

Transcranial magnetic stimulation (TMS) markers can help to shed some light on this investigation as they can index neuroplastic changes in the sensorimotor cortices, including the imbalance of inhibitory/excitatory circuits. Metrics from the TMS assessment paradigm, such as short intracortical inhibition (SICI) and intracortical facilitation (ICF), can indirectly measure the GABAergic and glutamatergic responses in the motor cortex, thus providing a surrogate metric of inhibitory/excitatory tonus [23]. It has been reported that SICI and ICF are altered compared to healthy subjects and other chronic conditions [4]; thus, TMS neurophysiological measures could help us to understand how the brain cortex can change in relation to clinical parameters. Indeed the relationship between these neurophysiological markers and the clinical and sensory characteristics of FMS patients is still unclear; the available evidence focuses more on distinguishing the FM subjects from the healthy population [24,25,26] rather than understanding the clinical correlates and validity of the TMS neurophysiological measures. Further exploration of these associations will help us to understand the role of TMS assessment as potential biomarkers of FMS phenotypes.

Therefore, we aimed to determine the clinical correlates of TMS neurophysiological measures in FMS patients. The motor cortex excitability—through single- and paired-pulse TMS protocols—and clinical and sensory characteristics of FMS patients were measured using standard clinical assessments and quantitative sensory testing (QST). Based on previous research [24,25,26], our hypothesis is that higher pain intensity, disease severity, and central and peripheral sensitization profiles would correlate less with primary motor cortex (M1) intracortical inhibitory tonus indexed by TMS neurophysiological measures.

## 2. Materials and Methods

### 2.1. Study Design and Participants

This study is a cross-sectional analysis that uses the baseline data (first visit with patients before any intervention occurred) from a randomized, double-blind clinical trial (Figure 1). The trial investigated the effects of tDCS in combination with aerobic exercise in fibromyalgia (FM) patients (NCT03371225) [27]. This study and all the assessments (including TMS) were approved by the IRB at the Mass General Brigham (protocol approval number: 2017P002524). All participants have given their informed consent.

We aimed to include adults aged between 18 and 65 years who have been diagnosed with FM according to the criteria outlined by the American College of Rheumatology (ACR) in 2010. These criteria were chosen because they were the most up-to-date recommendations at the time the protocol for the trial was written. These included: existing pain for more than six months with an average of at least 4 on a 0–10 Visual Analog Scale (VAS), and without any other comorbid chronic pain diagnosis; pain resistant to common analgesics and medication for chronic pain; and patients must have the ability to feel a sensation by stimulation and be able to provide informed consent. The diagnosis was confirmed by an experienced clinician (FF) and, in case of doubt, confirmed by using electronic medical records.

We excluded participants with clinically significant or unstable medical or psychiatric disorders; self-reported history of substance abuse within the past six months; previous important neurological history (e.g., traumatic brain injury) resulting in neurological deficits (self-reported); previous neurosurgical procedure with craniotomy; severe depression (with a score of >30 on the Beck Depression Inventory); current use of an opiate in large doses; pregnancy (as the risk of tDCS in pregnant population has not been assessed yet); increased risk for exercise as defined as not fulfilling the criteria outlined by the American College of Sports Medicine (ACSM) and not cleared by a licensed physician.

### 2.2. Clinical Variables

We performed a demographics survey and a brief clinical history (time since FM diagnosis, comorbidities, and medications). Moreover, we assessed several clinical domains, including pain intensity, depression, quality of life, health status, and sleep quality. The clinical variables were obtained from all of the participants. A complete description of scales is reported elsewhere [27], but the following is a brief summary of the assessments:

#### 2.2.1. The Revised Fibromyalgia Impact Questionnaire (FIQ)

This scale was used to assess the current health status of FM patients and includes three domains (function, overall impact and symptoms) with clinical and research relevance [28]. It consists of 21 items rated in scales from 0 to 10, with higher scores corresponding to greater dysfunction.

#### 2.2.2. The Quality of Life Scale (QOLS)

This scale is a questionnaire that includes 16 items evaluating areas in which chronic conditions could have an impact, using a 1 to 7 scale, with higher rates correlating with higher quality of life.

#### 2.2.3. Beck Depression Inventory-II (BDI-II)

This scale contains 21 items and is administered as a self-report of the symptoms experienced in the previous two weeks. The items are ranked from 0 to 3, with higher scores indicating more severe symptoms [29].

#### 2.2.4. Visual Analog Scale (VAS)

This tool presents numbers in a line with a range score of 0–10, which represent the continuum of the severity of the symptoms. Higher scores correlate with a higher severity of the symptoms. This tool was used to measure pain, anxiety, stress, and sleepiness.

#### 2.2.5. Pittsburgh Sleep Quality Index (PSQI)

This self-administered questionnaire aims to assess seven components of the patient’s sleep quality during the previous month [30]. It consists of 19 items, with higher scores denoting lower sleep quality.

#### 2.2.6. Patient-Reported Outcomes Measurement Information System (PROMIS)

This tool comprises a system of measures that are self-reported by the patients in relevant health outcomes and was used to measure pain, fatigue, and anxiety [31].

#### 2.2.7. Brief Pain Inventory (BPI)

This questionnaire collects the patient’s rating with respect to pain severity and how it impairs their daily functioning [32]. The short form was used, which contained 9 items and respondents were asked to give their answers on a scale from 0 to 10, with higher scores indicating greater severity. For this study, BPI was selected as the standardized measure of pain intensity and interference.

#### 2.2.8. Heart Rate

Heart rate was assessed using a wearable (Polar H01) during a thirty-minute walk on a treadmill at a speed chosen by the participant. Heart rate was measured on the same assessment day, usually between 9:00 am and 14:00 pm. The patients were asked not to consume any tea, coffee, or alcohol on the examination day. We measured three variables: Baseline heart rate, max heart rate, and the heart rate difference (max–baseline).

### 2.3. Quantitative Sensory Testing (QST) Profile

The sensory profiles of FM patients were assessed using a QST approach with heat stimulus. The variables measured were as follows: static QST (pain-60 threshold), dynamic QST (temporal slow pain summation [TSPS], and conditioned pain modulation [CPM]). For all protocols, TSA-II Stimulator (Medoc Advanced Medical Systems, Ramat Yishai, Israel) was located on the right proximal volar forearm using a 30 mm × 30 mm embedded heat pain (HP) thermode.

We first trained subjects to determine the pain-60 temperature threshold, i.e., the temperature that induced a painful sensation at a magnitude of 60 on a 0–100 numerical pain scale (NPS). We delivered three short heat stimuli (43, 44, and 45 °C), each lasting 7 s. Participants were asked to report their subjective levels of pain intensity using a VAS ranging from 0 (“no pain”) to 100 (“the worst pain imaginable”). If the three temperatures (43, 44, and 45 °C) were unable to achieve the pain-60, additional stimuli at 46, 47, and 48 °C were delivered until the desired pain level (60/100) was reached. In the unlikely event that none of those temperatures elicited pain-60, we considered it to be 48 °C.

For the TSPS protocol [33], the pulses were delivered with rise/fall of 1–2 s to reach the peak temperatures (participant’s pain-60), with a rate of change of 8 °C per second and a delta of 7 °C and a plateau of 0.7 s. They received one sequence of 15 repetitive heat stimuli at 0.4 Hz, and pain ratings were asked after the 1st and 15th stimuli. The TSPS was calculated as the difference between those ratings (after the 15th minus after the 1st). Positive values represent higher pain summation.

For the CPM protocol [12], after determining the pain-60, we applied the temperature for 30 s (test-stimulus), and subjects were asked to rate the pain level three times (after 10 s, 20 s, and 30 s). After we completed the test-stimulus, we waited 5 min to begin the conditioned-stimulus, where the left hand was immersed in a water bath at 10–12 °C for 30 s. Then, the same pain-60 temperature test was applied to the right forearm for 30 s, and the participant was asked to rate their pain another three times (at 10 s, 20 s, and 30 s). The CPM was calculated as the difference between the average pain rating for the test-stimulus minus the average pain rating in the conditioned-stimulus. Positive values represent larger CPM responses (pain inhibition).

### 2.4. Transcranial Magnetic Stimulation (TMS) Assessment

We measured motor cortex excitability using classical TMS with an EMG device. Specifically, a Magstim Rapid2 device with a figure-of-eight magnetic stimulator coil was placed on the right and left M1 (for all assessments), while the surface electromyogram was recorded from the contralateral first dorsal interosseous muscle. Single-pulse TMS was performed to acquire resting motor threshold (rMT) and motor-evoked potentials (MEPs), and paired-pulse TMS was used to measure short-interval cortical inhibition (SICI) and intracortical facilitation (ICF). As described by Rossini et al. [34], rMT was the lowest stimulus intensity to evoke an MEP of 50 μV in three of five trials in the relaxed muscle—recorded in both primary motor cortices. For the MEP, we adjusted the machine output for 120% of rMT to achieve a baseline MEP of 1 mV peak-to-peak amplitude, waited 7 s between the MEP trials, and recorded 10 MEPs before averaging their peaks.

A paired-pulse protocol was performed with a sub-threshold conditioning stimulus (80% of the rMT) followed by a supra-threshold test-stimulus of 120% of the rMT. In a randomized order, we included interstimulus intervals of 2 and 10 ms to elicit SICI and ICF, respectively. The randomization list of pulses included a MEP protocol (one pulse of 120% of the rMT) to reduce residual changes due to alternating pulses. Ten randomized stimuli were applied and averaged (peak-to-peak) from each condition (2 ms, 10 ms, and MEP), the SICI and ICF ratios were calculated as a function of the MEP condition using the following formulas, as in previous studies [12,35,36]:$$SICI_{\mathrm{ratio}}=\left(\frac{MEP\;condition}{MEP\;2ms}*100\right)$$
$$ICF_{\mathrm{ratio}}=\left(\frac{MEP\;baseline}{MEP\;after\;paired-pulse\;10\;ms}*100\right)$$

Additionally, we calculated the percentage of inhibition and facilitation by subtracting from 1 as follows:$$Inhibition_{\mathrm{percentage}}=\left(1-SICI\;ratio\right)$$
$$Facilitation_{\mathrm{percentage}}=\left(ICF\;ratio-1\right)$$

### 2.5. Statistical Analysis

The descriptive statistics were reported using either means with standard deviations (SD)—or medians with interquartile ranges—according to the data distribution. After checking assumptions for regression analysis with graphical methods (histograms and scatterplots), we performed a univariate linear regression analysis considering the TMS neurophysiological measures (SICI ratio, ICF ratio, rMT and MEP) as the dependent variable and the sociodemographic, clinical, and QST variables as the independent variables. We used a manual method to calculate the peak-to-peak MEP amplitude using the LabChart data analysis software (version 7.0; ADInstruments Ltd., Dunedin, New Zealand). Next, we performed a multivariate analysis, first including the output from the univariate models with the intention of diminishing the number of variables for the following analysis steps and reducing the chances of type 1. Thus, variables with a regression coefficient with significant *p*-value of <0.2 were included in an initial multivariate model. Then, using a backward elimination approach, the variable with the largest non-significant *p*-value was excluded from the model until a final model with variables with only significant coefficients (*p* < 0.05) was achieved. We identified potential confounders when a covariate changed the regression coefficient more than 20% compared to the unadjusted model. Therefore, statistically significant variables and potential confounders were included in the final multivariate models, always favoring the most parsimonious model. Multicollinearity was assessed for the final models via analysis of the variance inflation factors. Finally, we performed a sensitivity analysis, including interaction terms among clinical variables despite the literature suggesting potential effect modification (e.g., pain intensity*BMI), and tested a model with age, gender, fibromyalgia duration, and medication usage as predefined variables for adjustment. Statistical significance was set at *p* < 0.05, and all analyses were performed using a standard software package (Stata, version 15.0; StataCorp, College Station, TX, USA).

## 3. Results

### 3.1. Participants Characteristics

The clinical variables and demographic characteristics of the 58 subjects who participated in the experiment are summarized in Table 1.

### 3.2. QST and Cortical Excitability Profiles

The TMS and sensory findings can be seen in Table 2 and Table 3, respectively.

### 3.3. Regression Models

#### 3.3.1. SICI Models

In the univariate analysis (Table 4), SICI showed a significant negative association with fibromyalgia duration (*p* = 0.048)—the longer the disease duration, the higher the motor cortex inhibition. Our multivariate model showed a positive association of SICI with sleepiness (β = 7.504, 95% CI 2.32 to 12.687; *p* = 0.005) and the number of diseases (β = 3.99, 95% CI 0.456 to 7.516; *p* = 0.028), indicating that when the patient is sleepier and suffers from a higher number of comorbidities, there is less cortical inhibition (as indexed by SICI). A significant negative association was found between SICI and the duration of fibromyalgia (β = −2.532, 95% CI −3.990 to −1.074; *p* = 0.001) and anxiety (β = −7.236, 95% CI −12.15 to −2.322; *p*= 0.005)—the higher the duration of the disease (in months) and acute level of anxiety (VAS), the higher the intracortical inhibition. Finally, in our study, bilateral pain (similar pain intensity in both body sides) and heart rate difference (max–baseline) were outliers in the association between SICI and fibromyalgia duration. Further adjustments by age, gender, fibromyalgia duration, and medication usage did not change the results.

#### 3.3.2. ICF Models

From the univariate analyses, ICF negatively correlated with BMI and positively correlated with BPI pain intensity; specifically, the higher the ICF, the smaller the BMI and higher the pain intensity. Similarly, in the multivariate model, ICF showed a positive correlation between intracortical facilitation and BPI mean pain (β = 25.492, 95% CI 11.129 to 39.856; *p* = 0.001) and pain-60 (β = 22.482, 95% CI 5.472 to 39.491; *p* = 0.011), as well as a negative correlation with BMI (β = −4.067, 95% CI −7.018 to −1.117; *p* = 0.008). As in the ICF model, bilateral pain (similar pain intensity in both body sides) and heart rate difference (max–baseline) were outliers in the association between ICF and BPI pain intensity. Moreover, we performed a sensitivity analysis that included an interaction term between BPI pain intensity and BMI to explore whether BMI modified the association between pain intensity and ICF (Table 5). A statistically significant interaction (β = −2.069, 95% CI −4.103 to −0.034; *p* = 0.046) was found, meaning that the higher the BMI, the weaker the association between pain intensity and intracortical facilitation. Further adjustments by age, gender, fibromyalgia duration, and medication usage did not change the results.

#### 3.3.3. rMT Models

From the univariate analysis (Table 6), significant findings were made in relation to age (positive relationship, *p* = 0.007), alcohol (negative relationship, ep = 0.028), and maximum heart rate (negative relationship, *p* = 0.007). The multivariate analysis showed only an inverse relationship between motor threshold with history of alcohol use (β= −7.511, 95% CI −13.066 to −1.955; *p* = 0.009) and maximum HR (β = −0.546, 95% CI −0.894 to −0.199; *p* = 0.003)—a surrogate of adequate physical conditioning. Bilateral pain was the only outlier identified. Further adjustments by age, gender, fibromyalgia duration, and medication usage did not change the results.

#### 3.3.4. MEP Models

Motor evoked potential’s univariate analysis failed to yield significant associations. The multivariate analysis showed only a positive association between antidepressants (β= 0.218, 95% CI 0.003 to 0.434; *p* = 0.008) and MEP. Fibromyalgia duration, bilateral pain, and HR difference were found to be important outliers. Further adjustments by age, gender, and medication usage not pertaining to antidepressants did not change the results (Table 7).

## 4. Discussion

The current cross-sectional study investigated the clinical and QST correlates of TMS neurophysiological measures in FMS patients to improve the understanding of the clinical validity of cortical excitability as FMS biomarker. In summary, we found that higher SICI was associated with longer disease duration, less comorbidities, and lower levels of sleepiness, but it was not associated with pain intensity, disease severity, or QST alteration. Therefore, it seems SICI at baseline could be a marker of compensatory mechanisms. On the other hand, higher ICF was associated with higher pain intensity and lower body max index (BMI). The association between pain and ICF was negatively moderated by BMI (the strength was reduced within 1 unit of BMI). This highlights the fact that TMS assessment could represent a biomarker for pain intensity and BMI in patients with FMS. Furthermore, higher rMT is associated with the absence of alcohol ingestion and lower maximum heart rate during walking (surrogate of good physical conditioning), which is expected since corticospinal excitability is affected by psychoactive substances and exercise. Finally, no meaningful associated factor of MEP amplitude was found, which suggests that MEP amplitude is the least useful TMS metric for clinical application in FMS.

In contrast to our first hypothesis, no association between SICI and pain inhibition markers (QST profiles) or disease severity (pain intensity and FIQR) was found. It is well known that FM patients tend to have a higher SICI than those in the healthy population, which means they have a lower cortical inhibition [4,37,38]—probably mediated by dysfunction in the excitatory/inhibitory system—in which up-regulation in GABA-A receptors occurs [39]. However, according to our new findings and in alignment with findings from our previous studies on phantom limb pain [35] and FMS [12], ICI is a possible marker of adaptive or maladaptive compensation rather than a surrogate of clinical characteristics. A higher inhibition was associated with a higher duration of the disease, which suggests that SICI can be interpreted as a marker of adaptation to the chronification of FMS and potentially a measurement of prognosis and treatment response [4].

Similarly, SICI was associated with sleepiness levels and the number of comorbidities. How brain activation is changed to induce sleep is poorly understood. Although the circadian rhythm functions as an indicator of the time the brain is prepared to sleep, the mechanisms of change are still being studied. Evidence has shown an arousal inhibitory mechanism [40], which suggests an inhibitory influence from some part of the brain that initially influences the thalamus and then the cortex [41]. Consequently, our findings acknowledge these discoveries and propose a direct interference between sleep and intracortical inhibition. Additionally, previous data on a variety of diseases [42,43,44], mainly including but not limited to chronic pain conditions, have indicated an association between the disease and an imbalance in the intracortical inhibitory tonus in the motor cortex. Therefore, the overlapping of comorbidities can complicate these patterns. That said, it is our interpretation that inhibitory networks could be affected by health status and multimorbidity patterns (less physiological reserve). Taken together, these findings reaffirm the role of SICI as a marker of compensatory mechanisms in FMS (adaptation to chronification and comorbidities).

We found that high ICF correlates with high pain intensity but not SICI. SICI and ICF have been recognized as markers of the reorganization of sensorimotor pathways in chronic pain states; however, their roles seem to be associated with distinct facets of chronic pain since both markers are not always correlated [45,46,47]. Previous evidence has suggested that ICF is modified by GABA-meditated inhibition that could be independent of SICI changes (predominant GABA-A networks). Thus, the reduction in other types of cortical inhibition would lead to an increase in facilitation indexed by ICF [48,49], which could explain the high pain levels. These results are aligned with our previous study regarding phantom limb pain, which found that pain intensity was also associated to ICF, but not SICI [35]. We believe our findings can help to strengthen the evidence regarding the association between ICF as marker of disease severity in painful conditions.

Another interesting finding was the relationship between ICF and BMI. High rates of BMI are commonly associated with FMS and increased cardiovascular risks. However, no previous studies have explored the association of motor cortex excitability and body composition in fibromyalgia patients. It has been shown in animal studies that induced obesity is correlated with significant functional and structural changes in some areas of the brain and synaptic dysfunction, including glutamatergic neurons [50,51]. We hypothesize that similar remodeling due to inflammation could explain our findings that FMS patients with higher BMI have less motor cortex facilitation, which depends on glutamatergic activation. A distinct explanation could be due to the common association between higher BMI and lower rates of exercise, which could have negative effects on cortical excitability [52,53]. Further studies need to account for physical condition, physical activity, and inflammation to understand the influence of BMI on intracortical excitability.

Our study has some strengths and limitations. Firstly, unlike most previous studies that compared fibromyalgia patients with healthy controls, the novel aspect of our study is the comparison between TMS neurophysiological measures and several clinical and QST variables within a FMS population, which provided us with a greater understanding of the clinical validity of TMS neurophysiological measures. Regarding the limitations of our study, the number of participants was small, jeopardizing the statistical power and precision. Additionally, our study is cross-sectional, which is a static representation of reality, meaning further longitudinal research in the field is needed to confirm causal relationships. Moreover, certain variables were not measured in our study, including the MEP input-output recruitment curve, MEP latency, and the short afferent cortical inhibition (SAI). Future studies could explore these additional measures. Furthermore, we used the ACR 2010 diagnostic criteria to define FMS because the original RCT was approved and started before the new diagnostic criteria were released. However, we anticipate that differences in relation to sensitivity and specificity will not induce bias in our results since we are interested in broad FMS patients. Finally, participants that fulfilled the inclusion criteria for an ongoing clinical trial, including been able to engage in moderate exercise, were enrolled. This could induce selection bias and reduce the external validity of the results. We adjusted for a surrogate of physical conditioning (heart rate relative change) to diminish outliers in our analysis, but future studies including more diverse samples are needed.

## 5. Conclusions

TMS Paired-pulse protocols seem to have potential as phenotyping biomarkers in FMS. Specifically, our results suggest that SICI and ICF metrics correlate with disease compensation mechanisms and clinical characteristics (pain perception levels and BMI), respectively. Interestingly, BMI could moderate the relationship between pain and ICF, which deserves further study to explore its mechanisms. The clinical translation of TMS paired-pulse protocols represents an opportunity for a mechanistic understanding of FMS and the future development of precision treatments.

## Figures and Tables

**Figure 1 biomedicines-11-01543-f001:**
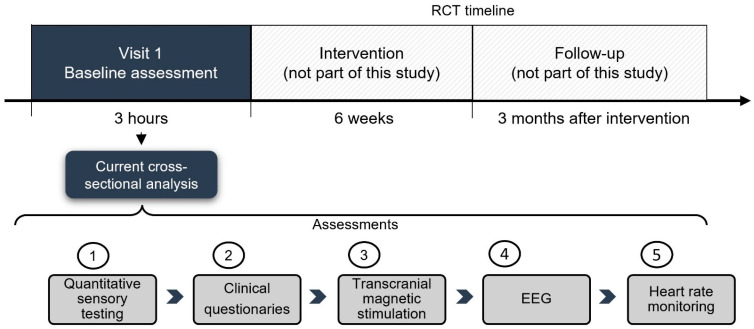
Timeline of the main study and cross-sectional analysis.

**Table 1 biomedicines-11-01543-t001:** Subjects demography data (*n* = 58).

Measurements	Mean ± SD or %
Age	47.62 ± 11.52
Gender (female, %)	50 (86.21%)
BMI	28.83 ± 8.13
Race (white, %)	44 (75.86%)
Heart rate	174.66 ± 8.06
Predominant painful side (right, %)	14 (24.14%)
Hand dominance (right, %)	52 (89.66%)
Duration of fibromyalgia (months)	10.94 ± 8.30
Education level (above high-school, %)	51 (87.93%)
Number of non pain-related comorbidities	6.97 ± 3.43
Smoking (yes, %)	11 (18.96%)
Alcohol (yes, %)	26 (44.83%)
Pain level (VAS)	5.76 ± 1.91
Anxiety level (VAS)	4.19 ± 2.57
Stress level (VAS)	5.10 ± 2.93
Depression level (VAS)	3.57 ± 2.85
Sleepiness level (VAS)	5.97 ± 2.66
Fatigue level (PROMIS)	3.84 ± 0.73
BDI total score	15.94 ± 8.99
Quality of life score	68.55 ± 14.70
FIQR	53.11 ± 19.62
PSQI	12.57 ± 4.47

BMI: body mass index; Number of non-pain related comorbidities: number of diseases that the patient has besides Fibromyalgia which are not other chronic pain conditions (i.e., sleep disturbance, mild depression, mild anxiety, etc.); VAS: visual analog scale; PROMIS: patient-reported outcomes measurement information system; BDI: back depression scale; FIQR: fibromyalgia impact questionnaire; PSQI: Pittsburgh sleep quality index.

**Table 2 biomedicines-11-01543-t002:** TMS findings (*n* = 58).

Measurements	Mean ± SD
MEP (mV)	0.98 ± 0.37
SICI ratio (%)	53.52 ± 46.38 (46% of inhibition)
ICF ratio (%)	177.63 ± 97.99 (77% of facilitation)
rMT (µV)	49.83 ± 10.23

MEP: motor-evoked-potential; SICI: short intracortical inhibition; ICF: intracortical facilitation; rMT: resting motor threshold; mV: millivolts; µV: microvolts.

**Table 3 biomedicines-11-01543-t003:** QST findings (*n* = 58).

Measurements	Mean ± SD
CPM response (VAS difference)	−0.859 ± 1.85
Pain-60 (°C)	46.50 ± 1.54
TSPS (VAS difference)	−0.36 ± 1.62

CPM: conditioned pain modulation; Pain-60: pain-60 test temperature; TSPS: temporal slow pain summation.

**Table 4 biomedicines-11-01543-t004:** Univariate and multivariate analysis for SICI.

Variable	ß-Coefficient	Unadjusted *p*-Value	95% Confidence Interval
Age	−0.770	0.153	(−1.824; 0.292)
Gender	17.640	0.323	(−17.98; 52.8)
BMI	0.160	0.832	(−1.362; 1.692)
Ethnicity (Non-white)	−13.940	0.332	(−42.59; 14.442)
Bilateral pain	−22.050	0.07	(−46.02; 1.803)
Bilateral hand dominance	−9.650	0.775	(−77.092; 57.721)
Duration of fibromyalgia	−1.460	0.048 *	(−2.90; −0.008)
Level of education (as ordinal)	−27.430	0.137	(−64.94; 9.14)
Number of diseases	0.680	0.707	(−2.94; 4.297)
Smoking	3.930	0.803	(−27.484; 35.274)
Alcohol use	−21.230	0.083	(−45.374; 2.80)
Anxiety (VAS)	−2.410	0.316	(−7.174; 2.383)
Depression (VAS)	1.960	0.369	(−2.345; 6.303)
Stress (VAS)	1.000	0.647	(−3.592; 4.992)
Sleepiness (VAS)	4.320	0.061	(−0.189; 8.844)
BDI total score	−0.33	0.63	(−1.711; 1.044)
BPI mean pain	3.250	0.4	(−4.424; 10.931)
BPI interference	0.480	0.873	(−5.54; 6.504)
BPI total score	1.800	0.628	(−5.61; 9.213)
Quality of life score	−0.100	0.807	(−0.945; 0.742)
QST pain 60 (°C)	2.388	0.583	(−6.276; 11.05)
Cpm response	−1.030	0.764	(−7.86; 5.807)
TSPS	−0.400	0.912	(−7.618; 6.822)
PSQI	1.060	0.448	(−1.697; 3.825)
FIQR	0.151	0.635	(−0.477; −0.785)
Antidepressant use	−7.370	0.569	(−33.066; 18.454)
Antiepileptics use	8.760	0.538	(−19.591; 37.012)
Baseline heart rate	0.580	0.269	(−0.458; 1.613)
Max heart rate	1.090	0.153	(−0.418; 2.606)
HR difference (max–baseline)	−0.060	0.901	(−1.025; 0.904)
HR difference (%)	0.571	0.972	(−31.501; 32.644)
Pain intensity (VAS)	5.524	0.125	(−1.586; 12.634)
Anxiety (PROMIS)	−2.662	0.535	(−9.914; 18.414)
Fatigue (PROMIS)	−1.798	0.581	(0.624; 33.556)
**Multivariate Analysis**	**ß-Coefficient**	**Adjusted *p*-Value**	**95% Confidence Interval**
Duration of fibromyalgia	−2.532	0.001 **	(−3.990; −1.074)
Sleepiness (VAS)	7.504	0.005 **	(2.320; 12.687)
Anxiety (VAS)	−7.236	0.005 **	(−12.150; −2.322)
Number of diseases	3.99	0.028 *	(0.456; 7.516)
Bilateral pain	−19.679	0.096	(−42.945; 3.586)
HR difference	−0.414	0.349	(−1.292; 0.465)

BMI: body mass index; Number of diseases: number of diseases that the patient has besides Fibromyalgia; VAS: visual analog scale; PROMIS: patient-reported outcomes measurement information system; BDI: back depression scale; FIQR: fibromyalgia impact questionnaire; PSQI: Pittsburgh sleep quality index CPM: conditioned pain modulation QST P60: quantitative sensory testing (pain-60 test temperature; TSPS: temporal slow pain summation. * *p* < 0.05; ** *p* ≤ 0.001.

**Table 5 biomedicines-11-01543-t005:** Univariate and multivariate analysis for ICF.

Variable	ß-Coefficient	Unadjusted *p*-Value	95% Confidence Interval
Age	−0.67	0.555	(−2.945; 1.595)
Gender	−15.63	0.679	(−92.753; 57.795)
BMI	−4.12	0.009 *	(−7.132; −1.059)
Ethnicity (non-white)	−6.85	0.822	(−68.646; 52.83)
Bilateral pain	9.51	0.716	(−42.971; 61.005)
Bilateral hand dominance	−24.02	0.737	(−166.664; 118.098)
Duration of fibromyalgia	−2.61	0.095	(−5.666; −0.499)
Level of education (as ordinal)	10.22	0.799	(−69.274; 90.291)
Number of diseases	−5.013	0.188	(−12.567; 2.498)
Smoking	29	0.382	(−37.2; 94.585)
Alcohol use	−10.21	0.697	(−62.886; 41.546)
Anxiety (VAS)	2.53	0.62	(−7.485; 12.84)
Depression (VAS)	4.55	0.323	(−4.404; 13.833)
Stress (VAS)	0.77	0.868	(−7.647; 10.296)
Sleepiness (VAS)	3.85	0.435	(−5.835; 13.755)
BDI total score	−1.26	0.386	(−4.16; 1.635)
BPI mean pain	22.24	0.005 *	(7.038; 37.441)
BPI interference	4.61	0.469	(−8.059; 17.276)
BPI mean total	13.14	0.086	(−1.926; 28.205)
Quality of life	−1.35	0.127	(−3.099; 0.395)
Pain-60	14.45	0.112	(−3.491; 32.394)
CPM response	5.51	0.403	(−7.599; 18.62)
TSPS	11.36	0.134	(−3.596; 26.307)
PSQI	−3.37	0.25	(−9.095; 2.503)
FIQR	0.5	0.451	(−0.8; 1.859)
Antidepressants use	−13.95	0.61	(−67.916; 41.034)
Antiepileptics use	9.76	0.745	(−50.58; 69.346)
Baseline heart rate	−0.798	0.471	(−3.001; 1.405)
Max heart rate	0.963	0.554	(−2.28; 4.208)
HR difference (max–baseline)	1.06	0.3	(−0.964; 3.074)
Pain intensity (VAS)	12.25	0.099	(−2.372; 26.889)
Anxiety (PROMIS)	14.86	0.098	(−23.777; 36.179)
Fatigue (PROMIS)	5.281	0.442	(−33.086; 39.114)
**Multivariate Analysis without Interaction**	**ß-Coefficient**	**Adjusted *p*-Value**	**95% Confidence Interval**
BMI	−4.067	0.008 **	(−7.018; −1.117)
BPI mean pain	25.492	0.001 **	(11.129; 39.856)
Pain-60	22.482	0.011 *	(5.472; 39.491)
Bilateral pain	18.38	0.413	(−26.327; 63.086)
HR difference	−0.591	0.547	(−2.568; 1.375)
**Multivariate Analysis with Interaction Term**	**ß-Coefficient**	**Adjusted *p*-Value**	**95% Confidence Interval**
Interaction between BPI mean pain and BMI	−2.069	0.046 *	(−4.103; −0.034)

BMI: body mass index; Number of diseases: number of diseases that the patient has besides Fibromyalgia; VAS: visual analog scale; PROMIS: patient-reported outcomes measurement information system; BDI: back depression scale; FIQR: fibromyalgia impact questionnaire; PSQI: Pittsburgh sleep quality index CPM: conditioned pain modulation; Pain-60: pain-60 test temperature; TSPS: temporal slow pain summation. * *p* < 0.05; ** *p* ≤ 0.001.

**Table 6 biomedicines-11-01543-t006:** Univariate and multivariate analysis for rMT.

Variable	ß-Coefficient	Unadjusted *p*-Value	95% Confidence Interval
Age	0.350	0.007 *	(0.976; 0.601)
Gender	5.490	0.227	(−3.511; 14.493)
BMI	0.040	0.824	(−0.341; 0.426)
Ethnicity	−0.990	0.773	(−7.803; 5.830)
Bilateral pain	1.330	0.661	(−4.687; 7.344)
Bilateral Hand dominance	0.700	0.639	(−13.210; 21.356)
Duration of fibromyalgia	0.200	0.273	(−0.163; 0.566)
Level of education (as ordinal)	−3.390	0.434	(−12.018; 5.228)
Number of diseases	0.730	0.087	(−0.108; 1.567)
Smoking	4.000	0.272	(−3.218; 11.215)
Alcohol use	−6.610	0.028 *	(−12.478; −0.749)
Anxiety (VAS)	−0.530	0.347	(−1.656; −0.591)
Depression (VAS)	−0.340	0.525	(−1.401; 0.723)
Stress (VAS)	0.060	0.905	(−1.05; 1.042)
Sleepiness (VAS)	0.125	0.828	(−1.017; 1.267)
BDI total score	−0.139	0.415	(−0.476; 0.199)
BPI mean pain	0.890	0.343	(−0.967; 2.74)
BPI interference mean	−0.410	0.586	(−1.906; 1.086)
BPI mean total	0.110	0.898	(−1.669; 1.898)
Quality of life	−0.132	0.205	(−0.338; 0.742)
Pain-60	0.770	0.48	(−1.405; 2.952)
CPM response	0.720	0.417	(−1.051; 2.5)
TSPS	−0.070	0.936	(−1.893; 1.746)
PSQI	−0.190	0.571	(−0.879; 0.489)
FIQR	0.030	0.675	(−0.123; 0.189)
Antidepressants use	2.068	0.5	(−4.031; 8.167)
Antiepileptics use	0.700	0.838	(−6.122; 7.527)
Baseline Heart Rate	0.313	0.811	(−0.23; 0.293)
Max Heart rate	−0.50	0.007 *	(−0.859; −0.139)
HR difference (max–baseline)	−0.223	0.06	(−0.456; 0.009)
Pain intensity (VAS)	−0.393	0.632	(−2.028; 1.242)
Anxiety (PROMIS)	1.107	0.288	(−3.016; 3.68)
Fatigue (PROMIS)	0.552	0.486	(−5.207; 3.056)
**Multivariate Analysis**	**ß-Coefficient**	**Adjusted *p*-Value**	**95% Confidence Interval**
Alcohol use	−7.511	0.009 **	(−13.066; −1.955)
Max heart rate	−0.546	0.003 **	(−0.894; −0.199)
Bilateral pain	0.626	0.819	(−4.839; 6.091)

BMI: body mass index; Number of diseases: number of diseases that the patient has besides Fibromyalgia; VAS: visual analog scale; PROMIS: patient-reported outcomes measurement information system; BDI: back depression scale; FIQR: fibromyalgia impact questionnaire; PSQI: Pittsburgh sleep quality index CPM: conditioned pain modulation; Pain-60: pain-60 test temperature; TSPS: temporal slow pain summation. * *p* < 0.05; ** *p* ≤ 0.001.

**Table 7 biomedicines-11-01543-t007:** Univariate and multivariate analysis for MEP.

Variable	ß-Coefficient	Unadjusted *p*-Value	95% Confidence Interval
Age	0.00	0.907	(−0.008; 0.009)
Gender	−0.06	0.650	(−0.412; 0.159)
BMI	0.006	0.321	(−0.005; 0.189)
Ethnicity	0.01	0.920	(−0.255; 0.208)
Bilateral pain	−0.03	0.775	(−0.242; 0.154)
Bilateral hand dominance	0.019	0.944	(−0.533; 0.553)
Duration of fibromyalgia	−0.001	0.830	(−0.012; 0.012)
Level of education (as ordinal)	−0.16	0.281	(−0.352; 0.151)
Number of diseases	0.001	0.970	(−0.029; 0.029)
Smoking	−0.06	0.631	(−0.322; −0.182)
Alcohol use	0.022	0.819	(−0.192; 0.206)
Anxiety (VAS)	0.013	0.508	(−0.021; 0.056)
Depression (VAS)	0.002	0.920	(−0.027; 0.043)
Stress (VAS)	−0.001	0.941	(−0.028; 0.039)
Sleepiness (VAS)	−0.011	0.549	(−0.045; 0.03)
BDI total score	0.005	0.343	(−0.004; 0.018)
BPI mean pain	0.008	0.789	(−0.054; 0.071)
BPI interference mean	0.011	0.637	(−0.037; 0.06)
BPI mean total	0.011	0.707	(−0.048; 0.071)
Quality of life	0.002	0.535	(−0.005; 0.009)
Pain-60	−0.01	0.781	(−0.08; 0.06)
CPM response	0.002	0.939	(−0.053; 0.057)
TSPS	−0.005	0.845	(−0.063; 0.052)
PSQI	−0.007	0.542	(−0.027; 0.018)
FIQR	0	0.951	(−0.004; 0.006)
Antidepressants use	0.17	0.088	(−0.012; 0.39)
Antiepileptics use	−0.12	0.285	(−0.358; 0.093)
Baseline heart rate	−0.002	0.632	(−0.010; 0.006)
Max heart rate	−0.001	0.916	(−0.013; 0.02)
HR difference (max–baseline)	0.001	0.708	(−0.006; 0.009)
Pain intensity (VAS)	0.023	0.418	(−0.034; 0.08)
Anxiety (PROMIS)	0.006	0.871	(−0.084; 0.144)
Fatigue (PROMIS)	−0.009	0.732	(−0.189; 0.084)
**Multivariate Analysis**	**ß-Coefficient**	**Adjusted *p*-Value**	**95% Confidence Interval**
Antidepressants use	0.218	0.047 *	(0.003; 0.434)
Duration of fibromyalgia	−0.005	0.512	(−0.018; 0.009)
Bilateral pain	−0.818	0.45	(−0.297; 0.134)
HR difference	0.0003	0.953	(0.133; 1.734)

BMI: body mass index; Number of diseases: number of diseases that the patient has besides Fibromyalgia; VAS: visual analog scale; PROMIS: patient-reported outcomes measurement information system; BDI: back depression scale; FIQR: fibromyalgia impact questionnaire; PSQI: Pittsburgh sleep quality index CPM: conditioned pain modulation; Pain-60: pain-60 test temperature; TSPS: temporal slow pain summation. * *p* < 0.05.

## Data Availability

The data that support the findings of this study are available on request from the corresponding author (F.F.).

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
