# Peer review of "Motor Cortex Inhibition and Facilitation Correlates with Fibromyalgia Compensatory Mechanisms and Pain: A Cross-Sectional Study"

_biomedicines, 2023, doi:10.3390/biomedicines11061543_

Round 1

Reviewer 1 Report

Line 48: In their study they included adults in the age range of 18 to 65 years. You say that the study carried out is a cross-sectional analysis. But there are many expensive tests that are carried out on participants of different ages. But they don't put the procedure on if they were scheduled for several days or they did everything in the same day because all the tests that you evaluate can cause significant fatigue in these patients.

Line 27: 8) Heart rate was assessed using a 26 wearable (Polar H01) during a walk of 30 min on a treadmill at a speed chosen by the 27 participant.

It can lead to bias. Perhaps stratify speeds at different levels to have this variable more controlled.

Table 1 details the number of comorbidities, however one of the inclusion criteria states "without any other comorbid chronic pain diagnosis" this may be misleading or explain it better.

Author Response

Reviewer 1

Line 48: In their study they included adults in the age range of 18 to 65 years. You say that the study carried out is a cross-sectional analysis. But there are many expensive tests that are carried out on participants of different ages. But they don't put the procedure on if they were scheduled for several days or they did everything in the same day because all the tests that you evaluate can cause significant fatigue in these patients.

Answer: Thank you for suggesting this clarification. The age range of the included participants (47.62 ± SD 11.52) is between 18 to 65 years, as mentioned in the protocol. This cross-sectional study was performed in the context of a randomized controlled trial (which is cited in the methods section). We are using the data from the baseline visit, a 3-hour in-person session with breaks. The order of assessments during the visit was heart rate monitor during aerobic exercise, quantitative sensory testing, questionaries, transcranial magnetic stimulation, and electroencephalography. We have clarified these details in the methods section.

Line 27: 8) Heart rate was assessed using a wearable (Polar H01) during a walk of 30 min on a treadmill at a speed chosen by the participant. It can lead to bias. Perhaps stratify speeds at different levels to have this variable more controlled.

Answer: Thank you for your comment. As mentioned previously, the heart rate measure was part of a personalized aerobic exercise protocol; therefore, we could not record or control the speed. However, since we are using an average heart rate metric only as a covariate in the analysis to account for physical conditioning, we believe any potential biases are minimized (no inferential interpretation of the heart rate was performed).

  • Table 1 details the number of comorbidities, however one of the inclusion criteria states "without any other comorbid chronic pain diagnosis" this may be misleading or explain it better.

Answer: Thank you for your feedback. Indeed, the sample did not have a comorbid chronic pain diagnosis; however, they presented with other comorbidities, predominantly mental health comorbidities (sleep disturbance, mild depression, mild anxiety, etc.). We have clarified this in the description of table 1, using the term “non pain-related comorbidities.”

Reviewer 2 Report

Thank you for the opportunity of reviewing this manuscript. The topic is of interest as determining a biomarker related with fibromyalgia would be of some interest as could help to classify fibromyalgia patients from other chronic pain patients. Despite of this, it seems that this would not help to treat fibromyalgia, which is the main challenge nowadays. In this aim, authors performed a study with only one group (I really do not understand why they called it "Randomized double blind trial"). I recommend the authors to solve some style, content and design issues:

Title must determine the type of study. Please follow TIDIeR guides. 

Page 1, Line 43: please pay attention to reference style of the journal. Correlative references must be placed together ([1,2] instead of [1][2]). Check it in the whole text.

Page 2- line 43: how could this be a randomized clinical trial if you only have one group? What did you randomized?

Page 1, Line 44: please specify that subpopulation that present an 80% of fibromyalgia prevalence.

Page 2, Line 3: authors state that “the treatment remains challenging with limited therapeutic efficacy”, supporting this affirmation with a study about serotonin and noradrenaline reuptake inhibitors, which is only one type of treatment. Authors must include references about other approaches (exercise, physiotherapy, painkillers…) to support their affirmation.

Page 2, lines 8-9: please support this sentence with a reference.

Page 2, lines 19: please place each reference next to the term of reference so readers could identify mechanism and reference.

Page 2, line 49: authors must know that the 2010 diagnostic criteria are not the most updated. In this line, they must explain why did not use the most updated ones (2016). Who did apply the diagnostic criteria? Was the same person than the one performing the TMS metrics? Had her/him enough experience with the diagnostic criteria of fibromyalgia?

Page 3, line 12: what do authors mean by “FM diagnosis”? Is it the time since patients were diagnosed?

Page 3, line 16: complete description of scales is required as the manuscript should be enough descriptive by itself.

Page 4, line 24: please delete “our”

Page 4, line 33: did you really mean to say “median” or “mean”?

REFERENCES are not in the journal style.

Author Response

Reviewer 2

  • Thank you for the opportunity of reviewing this manuscript. The topic is of interest as determining a biomarker related to fibromyalgia would be of some interest as could help to classify fibromyalgia patients from other chronic pain patients. Despite of this, it seems that this would not help to treat fibromyalgia, which is the main challenge nowadays. In this aim, authors performed a study with only one group (I really do not understand why they called it "Randomized double-blind trial"). I recommend the authors to solve some style, content and design issues:

Answer: Thank you for your comments. We have clarified that this manuscript is a secondary analysis of the baseline data of an ongoing randomized clinical trial, therefore we did not include any intervention in the present study.

  • Title must determine the type of study. Please follow TIDIeR guides. 

Answer: Thank you, we modified the title accordingly: “Motor cortex inhibition and facilitation correlates with fibromyalgia compensatory mechanisms and pain: a cross-sectional study”

  • Page 1, Line 43: please pay attention to reference style of the journal. Correlative references must be placed together ([1,2] instead of [1][2]). Check it in the whole text.

Answer: Thank you for pointing that out. We have corrected the reference style according to the journal guidelines as suggested.  

  • Page 2- line 43: how could this be a randomized clinical trial if you only have one group? What did you randomized?

Answer: Thank you for your comments. We have clarified that this manuscript is a secondary analysis of the baseline data of an ongoing randomized clinical trial, therefore we did not include any intervention in the present study. We improved the writing in the 2.1 section: “The study is a cross-sectional analysis using the baseline data (first visit with patients before any intervention occurred) from a randomized, double-blind clinical trial.”

  • Page 1, Line 44: please specify that subpopulation that present an 80% of fibromyalgia prevalence.

Answer: Thank you for raising that point. We have added the information in that sentence as: FMS affects around 2% of the general population, however in patients with comorbidities it could be as higher as 4% in patients in hemodialysis or up to 80% in patients with Behcet syndrome [3].”

  • Page 2, Line 3: authors state that “the treatment remains challenging with limited therapeutic efficacy”, supporting this affirmation with a study about serotonin and noradrenaline reuptake inhibitors, which is only one type of treatment. Authors must include references about other approaches (exercise, physiotherapy, painkillers…) to support their affirmation.

Answer: Thank you for your feedback. We have added more references for other treatment modalities.

  • Page 2, lines 8-9: please support this sentence with a reference.

Answer: Thank you for your request. We added references to those lines accordingly.

  • Page 2, lines 19: please place each reference next to the term of reference so readers could identify mechanism and reference.

Answer: Thank you for your suggestion, we have rearranged the citations accordingly.

  • Page 2, line 49: authors must know that the 2010 diagnostic criteria are not the most updated. In this line, they must explain why did not use the most updated ones (2016). Who did apply the diagnostic criteria? Was the same person than the one performing the TMS metrics? Had her/him enough experience with the diagnostic criteria of fibromyalgia?

Answer: Thank you for raising those questions. The protocol for the trial was developed before the release of the 2016 criteria. The diagnosis was confirmed by an experienced (Principal investigator Prof. Felipe Fregni) clinician who has enough experience with the diagnostic criteria, and in case of doubt confirmed by electronic medical records. The person who performed the TMS measures was a different co-investigator (KP) who was properly certified for this task. We described these details in methods as follows: “We aimed to include adults in the age range of 18 to 65 years, with the diagnosis of FM according to the American College of Rheumatology (ACR) 2010 criteria. These criteria were chosen because they were the most updated recommendations at the time the protocol for the trial was written. These included:  existing pain for more than six months with an average of at least four on a 0-10 Visual Analog Scale (VAS), and without any other comorbid chronic pain diagnosis; pain resistant to common analgesics and medication for chronic pain; and patients must have the ability to feel a sensation by stimulation and be able to provide informed consent. The diagnosis was confirmed by an experienced clinician (FF), and in case of doubt confirmed by using electronic medical records.” Also, we added limitations that we use the ACR 2010 criteria, because our study was approved and started before the new criteria was released.  

  • Page 3, line 12: what do authors mean by “FM diagnosis”? Is it the time since patients were diagnosed?

Answer: Thank you for noticing that. Yes, we meant the “time since FM diagnosis”. We have corrected that sentence.

  • Page 3, line 16: complete description of scales is required as the manuscript should be enough descriptive by itself.

Answer: Thank you for raising that point. We added a description and citation to all the included  scales.

  • Page 4, line 24: please delete “our”

Answer: Thank you for your suggestion. We had deleted that word.

  • Page 4, line 33: did you really mean to say “median” or “mean”?

Answer: Thank you for your question. We have rephrased that for the statement to be clearer: “The descriptive statistics were reported using either means with standard deviations (SD), or medians with interquartile ranges, according to the data distribution.”

  • REFERENCES are not in the journal style.

Answer: Thank you for your suggestion. We have corrected the “References” section according to the journal guidelines.

Reviewer 3 Report

-Please revise the abstract according to the journal guidelines with respect to the max number of words.

-Please revise the style of the referencing in the entire manuscript file according to journal guidelines (for example, raw 43 is written [1] [2], but should be [1,2], look raw 19, etc.)

-Please balance the usage of words“WE”, “we measured, , we hypothesized, we found, we delivered, we used, we waited, we included, we keep..in the entire manuscript. Currently, too many are used.

-Regarding the IRB and according to the provided title of the study, the study investigates the effects of tDCS. What about TMS that is used in the submitted paper?

-Please indicate the name of the TMS device, technical specifications, coil used, and stimulation parameters. Is it classical TMS with EMG device, line-navigated TMS, or e-field navigated TMS?

-The definition for determining the rMT should be 50 microvolts repeatable MEPs, not 100 microvolts as indicated in the submitted paper.

-Please use consistently rMT or MT in the paper, after introducing the full term and acronym, the usage of the acronym should be consistent in the entire manuscript text. Currently, it is not used consistently.

-raw 19, the space should be introduced for 10ms. Check also raw 22

-The data analysis should be part of the statistical analysis paragraph. Please introduce the method of how peak-to-peak MEP amplitude was estimated. Did the authors apply some automatic algorithm script? If yes, please state which method was used due to difficulties calculating MEP peak-to-peak amplitudes lower than 100 microvolts. Please check the paper from Šoda et al. 2021 (DOI: 10.1109/ACCESS.2020.3033075). It is visible that the authors used 120% of MT, but if the patient has a lower MT than 120% of MT sometimes varies. What is the lowest MT from your sample?

-The tables in the manuscript should be written and presented according to journal guidelines.

-Raw 12-13 “Thus, ICF at baseline seems to be a biomarker of FMS clinical characteristics.”Please explain to be more understandable.

- Raw 19-20 “Finally, no meaningful associated factor of MEP was found, which suggests

that MEP is the less useful TMS metric for clinical application in FMS”. The authors should elaborate on this more clearly. When presenting MEP, the reader should have more information about whether the authors are referring to MEP latency or MEP amplitude.

-Why was MEP latency not analyzed?

-Raw 23, “SICI larger” or higher or prolonged?

-It is suggested to clearly explain the term “inhibitory tonus” or rewrite it, not to mix it with muscle tonus in the whole manuscript.

-TMS “metrics” term should be replaced with TMS neurophysiological measures, for example.

-Is heart rate measured on the TMS examination day or? At what time? Did patients consummate tee or alcohol on the day of the TMS examination? The authors could present more clearly with some flowchart when different measures were assessed.

-The authors could include in the limitation (discussion) the fact that some neurophysiological measures were not investigated in the current sample, for example, MEP input output recruitment curve, MEP latency, and SAI (short afferent cortical inhibition)...

- “TMS markers” term (for example, raw 38) should be carefully used since TMS is the device, and specific neurophysiological measures are investigated. 

Author Response

Reviewer 3

  • Please revise the abstract according to the journal guidelines with respect to the max number of words.

Answer: Thank you for your recommendation. We have reduced the word count of the abstract according to the journal guidelines.

  • Please revise the style of the referencing in the entire manuscript file according to journal guidelines (for example, raw 43 is written [1] [2], but should be [1,2], look raw 19, etc.)

Answer: Thank you for pointing that out. We have corrected the reference style according to the journal guidelines as suggested.

  • Please balance the usage of words “WE”, “we measured, we hypothesized, we found, we delivered, we used, we waited, we included, we keep... in the entire manuscript. Currently, too many are used.

Answer: Thank you for the suggestion. The manuscript was corrected accordingly.

  • Regarding the IRB and according to the provided title of the study, the study investigates the effects of tDCS. What about TMS that is used in the submitted paper?

Answer: We have clarified this in the methods: “The study is a cross-sectional analysis using the baseline data (first visit with patients before any intervention occurred) from a randomized, double-blind clinical trial. The trial investigated the effects of tDCS in combination with aerobic exercise in fibromyalgia (FM) patients (NCT03371225)[24]. This study and all the assessments (including TMS) were approved by the IRB at the Mass General Brigham (protocol approval number: 2017P002524). All participants have given their informed consent.”

  • Please indicate the name of the TMS device, technical specifications, coil used, and stimulation parameters. Is it classical TMS with EMG device, line-navigated TMS, or e-field navigated TMS?

Answer: Thank you for raising those questions. We have added these details in the methods section: “We measured motor cortex excitability using classical TMS with an EMG device. Specifically, a Magstim Rapid2 device with a figure-of-eight magnetic stimulator coil was placed on the right and left M1 (for all assessments), while the surface electromyogram was recorded from the contralateral first dorsal interosseous muscle.”

  • The definition for determining the rMT should be 50 microvolts repeatable MEPs, not 100 microvolts as indicated in the submitted paper.

Answer: Thank you for noticing that error, it was corrected in the methods section.

  • Please use consistently rMT or MT in the paper, after introducing the full term and acronym, the usage of the acronym should be consistent in the entire manuscript text. Currently, it is not used consistently.

Answer: Thank you for pointing that out. We have now used the term rMT and corrected it throughout the manuscript.

  • raw 19, the space should be introduced for 10ms. Check also raw 22

Answer: Thank you for your suggestion. It was corrected in the text.

  • The data analysis should be part of the statistical analysis paragraph. Please introduce the method of how peak-to-peak MEP amplitude was estimated. Did the authors apply some automatic algorithm script? If yes, please state which method was used due to difficulties calculating MEP peak-to-peak amplitudes lower than 100 microvolts. Please check the paper from Šoda et al. 2021 (DOI: 10.1109/ACCESS.2020.3033075). It is visible that the authors used 120% of MT, but if the patient has a lower MT than 120% of MT sometimes varies. What is the lowest MT from your sample?

Answer: Thank you for your suggestions. We added this is methods: “We have used a manual method to calculate the peak-to-peak MEP amplitude using LabChart data analysis software (version 7.0; ADInstruments Ltd)”. About the second question, in our sample the MTs were high, specifically the lowest was 40%. Therefore using 120% of the MT is a stable measure to get a reliable MEP.

  • The tables in the manuscript should be written and presented according to journal guidelines.

Answer: Thank you for noticing that. We have corrected the tables style according to the journal guidelines as suggested.

  • Raw 12-13 “Thus, ICF at baseline seems to be a biomarker of FMS clinical characteristics. “Please explain to be more understandable.

Answer: Thank you for your suggestion. The sentence was corrected to: “This highlights the fact that TMS assessment could represent a biomarker for pain intensity and BMI in FMS.”

  • Raw 19-20 “Finally, no meaningful associated factor of MEP was found, which suggests that MEP is the less useful TMS metric for clinical application in FMS”. The authors should elaborate on this more clearly. When presenting MEP, the reader should have more information about whether the authors are referring to MEP latency or MEP amplitude.

Answer: Thank you for pointing that out. We have corrected that sentence into: “Finally, no meaningful associated factor of MEP amplitude was found, which suggests that MEP amplitude is the less useful TMS metric for clinical application in FMS.”

  • Why was MEP latency not analyzed?

Answer: Thank you for raising that question. Considering that our main hypothesis is related with intracortical inhibition and facilitation, we assumed that the corticospinal transmission of the motor pathway was not affected in patients with FMS. Therefore, changes in the latency were not expected. However, we agree that future studies should explore more TMS measures as potential biomarkers.

  • Raw 23, “SICI larger” or higher or prolonged?

Answer: Thank you for asking that question. We have corrected that phrase into: “It is well-known that FM patients tend to have a SICI higher than a healthy population, which means less cortical inhibition.”

  • It is suggested to clearly explain the term “inhibitory tonus” or rewrite it, not to mix it with muscle tonus in the whole manuscript.

Answer: Thank you, we have corrected it throughout the manuscript and replaced that phrase with: “Intracortical inhibitory tonus in the motor cortex”.

  • TMS “metrics” term should be replaced with TMS neurophysiological measures, for example.

Answer: Thank you for your suggestion, we have corrected the terminology and used the term “TMS neurophysiological measures” throughout the manuscript.

  • Is heart rate measured on the TMS examination day or? At what time? Did patients consummate tee or alcohol on the day of the TMS examination? The authors could present more clearly with some flowchart when different measures were assessed.

Answer: Thank you for suggesting this clarification. This cross-sectional study was performed in the context of a randomized controlled trial (which is cited in the methods section). We are using the data from the baseline visit, a 3-hour in-person session with breaks. The order of assessments during the visit were: quantitative sensory testing, questionaries, transcranial magnetic stimulation, and electroencephalography and then was heart rate monitor during aerobic exercise. A small chart of this process was added in Figure 1. We have clarified these details in the methods section: “Heart rate was measured on the same assessment day, usually between 9:00 am and 14:00 pm. The patients were asked not to take any tea, coffee, or alcohol on the examination day.”

  • The authors could include in the limitation (discussion) the fact that some neurophysiological measures were not investigated in the current sample, for example, MEP input output recruitment curve, MEP latency, and SAI (short afferent cortical inhibition)...

Answer: Thank you for your suggestion. We have added this point in the limitation section: “Moreover, certain variables were not measured in our study including the MEP input-output recruitment curve, MEP latency, and the short afferent cortical inhibition (SAI). Future studies could explore these additional measures.”

  • “TMS markers” term (for example, raw 38) should be carefully used since TMS is the device, and specific neurophysiological measures are investigated. 

Answer: Thank you for noticing that. We have clarified the terms throughout the manuscript and changed that phrase into “TMS neurophysiological measures”.

Round 2

Reviewer 2 Report

Thank you for the improvements performed.